# The impact of future time perspective on academic achievement: Mediating roles of academic burnout and engagement

Shuai Hong[1,2]*

1 School of Education, Shandong Women's University, Jinan, P.R. China, 2 School of Education, Faculty of Social Sciences and Leisure Management, Taylor's University Lakeside Campus, Subang Jaya, Malaysia

* shuai.hong@sdwu.edu.cn

**Data Availability Statement:** All relevant data are within the paper and its Supporting Information files.

**Funding:** The author(s) received no specific funding for this work.

## Abstract

Future time perspective is closely related to students' academic achievement, potentially affecting academic achievement through academic emotions. This study examines the relationship between future time perspective and academic achievement, exploring the mediating roles of academic burnout and academic engagement among 686 university students. Using a cross-sectional quantitative design, we investigated both the direct relationship between future time perspective and academic achievement, and the indirect effects mediated by burnout and engagement. Structural equation modeling revealed that future time perspective predicts academic achievement directly, and also indirectly through increasing academic engagement and reducing academic burnout. Notably, the mediating effect of academic burnout was found to be more significant than that of academic engagement. Overall, the results highlight the crucial role of future time perspective and its mediating mechanisms in promoting academic success, suggesting potential interventions to enhance students' future orientation and academic well-being.

## Introduction

Academic achievement is a critical benchmark within the education system, profoundly impacting individual students' future trajectories and serving as a key indicator of the overall quality of the education system [1–3]. It garners significant attention from various stakeholders, including parents, educators, educational institutions, and the students themselves. Understanding the factors influencing academic achievement is essential for enhancing educational outcomes. Past research has established a link with respect to future time perspective and academic achievement [4–8]. However, the mediating mechanisms underlying this relationship remain incompletely understood. Based on empirical evidence, there are interrelationships among future time perspective, academic burnout, academic engagement, and academic achievement [9, 10]. This highlights the need for in-depth exploration of whether time perspective influences academic achievement through the mediating variables of academic burnout and engagement. This study focuses on undergraduate students in early

**Competing interests:** The authors have declared that no competing interests exist.

adulthood, a crucial period for cultivating self-efficacy [11] and future planning [12, 13]. During this stage, the personal trait of future time perspective is expected to exert a greater influence on academic achievement than parental practices and support [14] and institutional support [15]. This study aims to examine the effect of future time perspective on academic achievement and to explore the potential mediating roles of academic burnout and engagement among undergraduate within the Chinese higher education system. The significance of this study lies in its potential to empower educators and policymakers to design targeted interventions that help students overcome academic challenges, enhance their academic performance, and improve their psychological well-being.

## Future time perspective and academic achievement

Future time perspective (FTP) is a personal psycho-temporal trait that has garnered significant attention in recent research due to its broad implications for motivation and behavior, especially in academic contexts. FTP entails a keen emphasis on future goals and their outcomes [5], relating to the long-term objectives that individuals aim to achieve [16–18]. Studies suggest that younger adults appears to have more extended future time perspective [19]. People with a more pronounced future time perspective are better equipped to engage in rational thinking and self-control during decision-making [20]. They are more likely to choose long-term options for greater benefits and adopt healthier lifestyles [21]. An example of a future-time perspective is, "When I want to achieve something, I set specific goals to reach it." Research among undergraduates has demonstrated that future time perspective is linked with adopting healthier behaviors [21] and can significantly predict self-reported health behaviors [22]. Additionally, there is evidence that conscientiousness, consideration of future consequences, self-reported weekly study hours, and a preference for consistency are all positively associated with future time perspective [17]. Furthermore, future time perspective enhances the prediction of health behaviors such as eating breakfast regularly, wearing a safety belt while driving, and engaging in regular exercise [22].

In the contexts of education, researches indicated that future time perspective had motivational functions, with evidence indicating that students with a strong FTP place high value on their academic goals, positively influencing academic achievement [4, 6, 7, 23]. Research by Loose and Vasquez-Echeverría demonstrated that among the five dimensions of time perspective, future time perspective was the sole predictor significantly associated with academic performance, while present and past perspectives showed no such association [24]. Similarly, studies by Mello and Worrell [25] and Phan [26] demonstrated that the future dimension of time perspective effectively predicted academic achievement, learning objectives, and strategies. Barnett et al. provided additional support for this finding, showing that future time perspective significantly predicted students' learning scores [5]. Empirical evidence also supports the notion that FTP is positively associated with academic delay of gratification, self-regulated learning [27], and learning autonomy [28].

In recent years, FTP has become a notable research focus within psychology due to its strong predictive value for academic and health-related outcomes. Its growing importance in academic studies highlights FTP as a critical factor for promoting student success and well-being in high-pressure academic environments [29], underscoring its relevance as a tool for understanding and improving academic achievement.

## The role of academic burnout and engagement

Understanding why traits predict outcomes necessitates research on mediating mechanisms. Negative and positive emotionality are among the mechanisms through which personality

processes unfold and exert influence [30]. Academic burnout and academic engagement, as two opposing aspects of academic well-being [31], are associated with time perspective, implying their impact on academic achievement. Students with a stronger future time perspective are more inclined to invest time and effort into their studies and achieve better academic outcomes [8, 27, 32]. FTP also negatively predicts procrastination [33], further supporting its role in promoting academic engagement and performance. Barnett et al. confirm that future time perspective is positively related to intended academic engagement and GPA, serving as a predictor for both [5].

While academic engagement is associated with better academic achievement [9], high levels of academic burnout increases the risk of various academic challenges such as internet addiction [34], dropping out of school [35], and poor academic performance [35, 36]. Academic burnout is a pervasive issue among university students, particularly in high-pressure academic environments like those in China. Research highlights a significant prevalence of this problem: a recent cross-sectional study involving 22,983 Chinese university students found that 59.9% experienced academic burnout [37]. Hong and Hanafi similarly reported an upward trend in academic burnout rates among Chinese university students, with prevalence increasing from 37.5% in 2007 to 61.63% in 2019 [38]. Other studies highlight that academic burnout, alongside internet addiction and loneliness, is a significant challenge among Chinses college students and is often linked with heightened stress levels [39]. Academic burnout can have profound effects on students' academic performance and achievement, underscoring the importance of fostering academic engagement as a protective factor and a means of enhancing students' academic success.

The relationship between academic burnout and engagement itself remains underexplored. Previous studies have suggested that these two constructs, though opposing, may be interconnected. For example, research has demonstrated that English learning burnout negatively affects the psychological resources required for engagement, with resilience and motivation mediating this relationship [40]. This suggests that while burnout and engagement are often viewed as separate variables, they may influence each other in a complex manner. Reducing academic burnout and fostering engagement can significantly improve students' well-being [31] and enhance academic achievement [41].

While substantial research supports the relationships among these variables, few studies have integrated all four into a single model to explore how academic burnout and engagement mediate these connections. One comparable study conducted by King examined time perspective as an antecedent, with academic engagement and disaffection serving as mediators, and academic achievement as the ultimate outcome [9]. However, this study focused on Filipino university students, primarily first-year undergraduates, and did not explore the role of burnout. More recently, Barnett et al. explored the correlation between future time perspective and GPA (Grade Point Average), mediated by intended academic engagement, but again, academic burnout was not included [5].

Combining Zimbardo's Time Perspective Theory (ZTPT) and Self-Determination Theory (SDT) provides a nuanced framework for understanding how Future Time Perspective (FTP) influences academic achievement through its effects on academic burnout and engagement. According to Zimbardo's theory, FTP is the tendency to prioritize future goals and consider long-term implications, prompting goal-oriented behaviours and long-term planning [18]. Individuals with a strong FTP prioritize actions that align with future success, fostering adaptive academic behaviours and effective resource allocation in pursuit of their goals [4]. SDT complements this perspective by emphasizing the role of basic psychological needs–specifically, competence, autonomy, and relatedness–in fostering intrinsic motivation and self-regulation [42]. FTP, as a future-focused motivational resource [4], facilitates the satisfaction of

competence and autonomy needs by helping students view academic tasks as meaningful steps toward future goals. Students with a high FTP are more likely to persist through academic challenges, motivated by the anticipated benefits of their efforts [27]. This forward-thinking approach reduce academic burnout by framing academic demands within a long-term, purposeful plan, rather than as immediate burdens. FTP also promotes academic engagement by focusing students on future rewards and encouraging sustained effort and interest in their studies. Aligning future goals with basic psychological needs fosters both academic engagement and resilience [40], equipping students to handle academic demands with reduced burnout and increased enthusiasm. Thus, FTP reduces academic burnout by conserving students' mental and emotional resources while enhancing academic engagement through active learning and goal-directed effort. Our model integrates ZTPT and SDT to highlight FTP as a crucial factor in reducing academic burnout and promoting academic engagement, ultimately contributing to academic achievement.

Overall, there remains a notable gap in our understanding of the mechanism through which future time perspective influences academic achievement. Identifying these mechanisms is crucial for comprehending the relation between future time perspective and academic achievement and for developing effective interventions to enhance students' academic success. In this study, we chose to focus on the direct and indirect effects of future time perspective (FTP) on academic achievement, with academic burnout and academic engagement as two distinct mediators. We did not explicitly model the relationship between burnout and engagement in our framework to avoid overcomplicating the model, but we recognize that these two factors may interact in ways suggested by previous research [43], which further highlights the potential for future research to explore this connection. By examining these pathways, this present study aims to fill this research gap by investigating the separate roles of academic burnout and engagement in the relationship between FTP and academic achievement.

## The current study

Considering the theories of time perspective and self-determination, the main goal of this study is to investigate the relationship between future time perspective and academic achievement among undergraduate students in China. Additionally, this study focuses on the potential mechanisms that explain this relationship. Given the theoretical expectations and empirical evidence linking future time perspective to academic achievement, academic burnout, and academic engagement [9, 10, 20, 44], another objective is to explore whether burnout and engagement mediate the relationship between future time perspective and academic achievement within a single model. Drawing on prior research, we hypothesized four specific relationships: (1) Future time perspective positively predicts academic achievement; (2) Future time perspective influences academic achievement through academic engagement, with engagement mediating this relationship; (3) Future time perspective influences academic achievement through academic burnout, with burnout mediating this relationship. The conceptual model is depicted in Fig 1.

## Methods

### Participants

753 undergraduate students were recruited from two comprehensive public universities to voluntarily participate in a study titled "Exploring the Relationship Between Time Perspective and Academic Achievement". After excluding incomplete surveys and those deemed to be inattentive responses, the final sample consisted of 686 undergraduate students (71.1% female; age range: 18–28; $M_{age}$ = 19.31 ± 1.48 years). The distribution across academic years was as

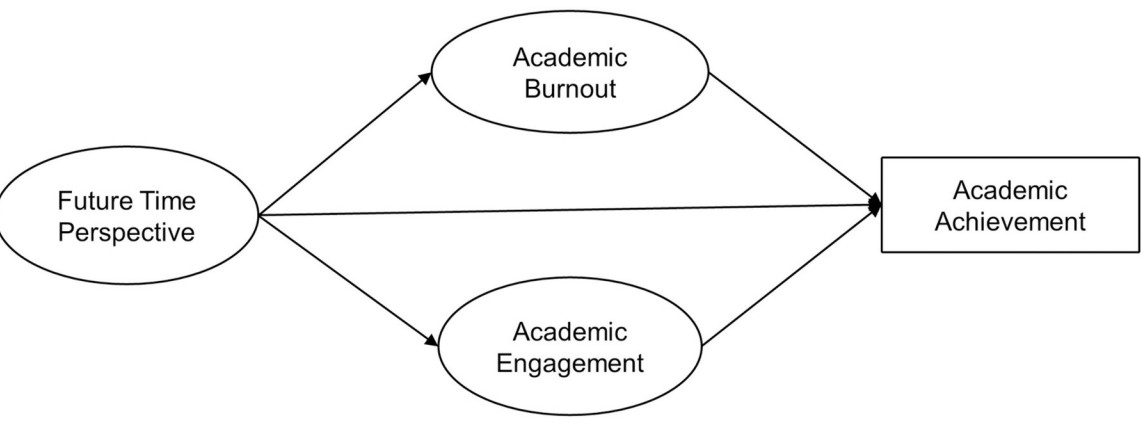

**Fig 1. The conceptual model.**

follows: freshman 255 (37.2%), sophomore 278 (40.5%), junior 83 (12.1%), senior 70 (10.2%). Urban students accounted for 243 (35.4%), while rural students comprised 442 (64.4%).

## Procedures

The survey was administered from October 9 to November 10, 2023, spanning the 7th to 11th weeks of the first semester of the 2023–2024 academic year. This study involved data collection from two sources at two different time points. In the initial phase, participants completed an online self-report survey assessing their future time perspective, academic burnout, academic engagement, and demographic information. The survey required approximately 20 minutes to complete. During the second phase, researcher obtained the participants' GPA scores for the current semester from the academic administration department. Although there was a one-month interval between the two data collection points, we treated the data as cross-sectional rather than longitudinal, given the short interval and the fact that the end-of-semester GPA reflects the academic status for that specific term.

All participants provided written informed consent. No compensation was provided for the participants. The study received ethical approval from the Taylor's University Human Ethics Committee, and was performed in accordance with the ethical standards as laid down in the 1964 Declaration of Helsinki and its later amendments or comparable ethical standards.

## Measures

**Future time perspective.** Future time perspective was measured using the Chinese version of the Zimbardo Time Perspective Inventory (ZTPI-Chinese, see S1 Appendix) [45]. The future time perspective subscale (5 items; e.g., "Meeting tomorrow's deadlines and doing other necessary work comes before tonight's play") assessed the extent to which focus is placed on future goals and future outcomes. Responses are recorded on a five-point scale, ranging from 1 (strongly disagree) to 5 (strongly agree). The items of this subscale were averaged to calculate the subscale score. The reliability and validity of the measure have been confirmed in the Chinese context [45]. In this study, the Cronbach's alpha coefficient was 0.75, indicating acceptable internal consistency. Confirmatory factor analysis (CFA) demonstrated acceptable construct validity for future time perspective measurement ($\chi^2$ = 15.56, df = 5, RMSEA = 0.06, CFI = 0.98, TLI = 0.97, SRMR = 0.02).

**Academic burnout.** Academic burnout was measured using the Learning Burnout of Undergraduate Scale (LBUS, see S2 Appendix) [46]. The LBUS was adapted based on

Maslach's burnout theory and the Maslach Burnout Inventory [47, 48], maintaining a three dimensional structure to measure the severity of academic burnout in university students. The LBUS consists of 20 items categorized into three dimensions: emotional exhaustion (8 items; e.g., "After a long day of studying, I feel completely worn out"), misbehavior (6 items; e.g., "I hardly ever plan or schedule my study time"), and inefficacy (6 items; e.g., "Mastering my course material comes pretty easily to me"). The inefficacy dimension's items are positively worded and require reverse scoring. Responses were recorded on a 5-point scale ranging from 1 (strongly disagree) to 5 (strongly agree), with higher scores indicating stronger endorsement of the item. Dimension scores are the average of the corresponding items, and the total LBUS score is the average of all items. In this study, Cronbach's alpha coefficients were 0.85 for exhaustion, 0.73 for inefficacy, and 0.78 for misbehavior, with an overall LBUS scale reliability of 0.90. Confirmatory factor analysis demonstrated acceptable construct validity with fit indices: $\chi^2$ (116) = 481.27, $p < .001$, CFI = .906, TLI = .889, RMSEA = .075, SRMR = .048.

**Academic engagement.** Students' academic engagement was measured using the Chinese adaptation of the Utrecht Work Engagement Scale for Student (UWES-S, see S3 Appendix) [49]. The scale includes three subscales with a total of 17 items: vitality (6 items; e.g., "I feel full of energy when studying or attending classes"), dedication (5 items; e.g., "I believe learning is valuable and meaningful"), and absorption (6 items; e.g., "When I'm studying, I get so focused that I lose track of what's happening around me"). Responses were rated on a 7-point scale, ranging from 1 (never) to 7 (always), with higher scores indicating greater academic engagement. In this study, Cronbach's alpha coefficients for the overall scale and its subscales ranged from 0.89 to 0.96.

Regarding structural validity, due to the high correlations among the three dimensions in this study ($r > 0.8$, $p < 0.01$), we follow Zhang et al. [50] and evaluated two competing models for academic engagement: Model 1, a single-factor model including only academic engagement, and Model 2, which retained the original three-factor structure of the UWES-S with vitality, dedication, and absorption as first-order latent variables, and academic engagement as a second-order latent variable. The CFA results indicated that the three-factor model exhibited an acceptable fit ($\chi^2$ (99) = 591.99, p < .001, CFI = .941, TLI = .929, RMSEA = .088, SRMR = .032). In contrast, the one-factor model demonstrated a better fit to the data ($\chi^2$ (99) = 583.974, p < .001, CFI = .942, TLI = .930, RMSEA = .087, SRMR = .032). Given the non-nested nature of these models, we selected Model 1, which better fits the data for this study.

Indicators of the reliability and validity of the above three measurement instruments can also be found in S4 Appendix.

**Academic achievement.** Academic achievement was assessed by the grade point average (GPA). Two months after the completion of the survey, i.e. at the end of the semester, the GPA of the participants was obtained by matching the student numbers provided in the survey with the records held by academic administration department. Then the corresponding GPAs for the participating students were extracted for analysis.

**Demographic variables.** We included age, gender, grade level, and place of birth as control variables in the demographic form.

## Data analysis

This study employed structural equation modeling (SEM) with Mplus 8.3 [51] to analyze the relationships between future time perspective, academic burnout, academic engagement, and academic achievement. Specifically, we first examined the total effect model of future time perspective on academic achievement. Then, the mediating effects of academic burnout and academic engagement in the relationship were tested. A 95% bias-corrected bootstrap method

was utilized to evaluate the significance of the total and indirect effects. An effect was considered significant if the 95% confidence interval (95% CI) exclude zero. The model fit was evaluated using several indices, including the Root Mean Square Error of Approximation (RMSEA), Comparative Fit Index (CFI), and Tucker-Lewis Index (TLI). The model was deemed to have acceptable fit if the RMSEA was below 0.08 and the CFI and TLI were above 0.90 [52].

## Results

### Descriptive statistics

The means, standard deviations, and correlations among future time perspective, academic burnout, academic engagement, and academic achievement are presented in Table 1. Future time perspective was positively correlated with both academic engagement and academic achievement ($r = 0.42, 0.39, p < 0.01$) and negatively correlated with academic burnout ($r = -0.53, p < 0.01$). Academic achievement showed a negative correlation with academic burnout ($r = -0.48, p < 0.01$) and a positive correlation with academic engagement ($r = 0.41, p < 0.01$). Additionally, academic burnout and academic engagement were negatively correlated ($r = -0.56, p < 0.01$).

### Mediating effects of academic burnout and engagement

First, we examined the total effect model of future time perspective (FTP) on academic achievement. The model fit indices met statistical standards, indicating a satisfactory fit for the total effect model ($\chi^2 = 20.23$, df = 9, $p < 0.05$; RMSEA = 0.043; CFI = 0.98; TLI = 0.97; SRMR = 0.022). FTP showed a positive total effect on academic achievement ($\beta = 0.38$, $p < 0.001$, 95% CI [0.28, 0.45]).

Next, we included academic burnout and academic engagement as mediating variables to investigate their roles in the relationship between FTP and academic achievement, using bias-corrected bootstrapping with 5000 resamples. The fit indices for the mediation model, which included both mediators, indicated an acceptable fit to the data ($\chi^2 = 975.38$, df = 246; RMSEA = 0.067; CFI = 0.881; TLI = 0.866; SRMR = 0.056). **Table 2** showed that academic burnout mediated the relationship between FTP and academic achievement (95% CI [0.16, 0.33]), as did academic engagement (95% CI [0.003, 0.07]). However, the mediation effect of

**Table 1. Descriptive statistics and correlations.**

| Variables | *M* | *SD* | 1 | 2 | 3 | 4 | 5 | 6 | 7 | 8 | 9 |
|---|---|---|---|---|---|---|---|---|---|---|---|
| 1 FTP | 3.68 | .62 | **.75** | | | | | | | | |
| 2 AB | 2.81 | .58 | -.53** | **.90** | | | | | | | |
| 3 Exhaustion | 2.83 | .74 | -.42** | .92** | **.86** | | | | | | |
| 4 Misbehavior | 2.84 | .65 | -.55** | .88** | .74** | **.78** | | | | | |
| 5 Inefficacy | 2.76 | .60 | -.44** | .77** | .55** | .56** | **.73** | | | | |
| 6 AE | 4.08 | .95 | .42** | -.56** | -.46** | -.55** | -.47** | **.96** | | | |
| 7 Vitality | 3.88 | 1.0 | .39** | -.54** | -.44** | -.54** | -.44** | .96** | **.91** | | |
| 8 Dedication | 4.40 | .99 | .43** | -.57** | -.49** | -.53** | -.46** | .93** | .82** | **.89** | |
| 9 Absorption | 3.94 | 1.02 | .35** | -.47** | -.37** | -.47** | -.41** | .95** | .89** | .80** | **.90** |
| 10 AA | 3.30 | .51 | .39** | -.48** | -.40** | -.46** | -.40** | .41** | .36** | .44** | .35** |

*Note*. The Cronbach's α of each variable was listed in the diagonal in boldface type; FTP = future time perspective; AB = academic burnout; AE = academic engagement; AA = academic achievement; $^*p < .05$; $^{**}p < .01$; $^{***}p < .001$.

**Table 2. Mediation analyses of the effects of academic burnout and engagement on future time perspective and academic achievement.**

| Model pathways | Standardized estimate | SE | p-value | 95% CI |
|---|---|---|---|---|
| **FTP to AA** | | | | |
| Total effect | .39[a] | .04 | < .001 | [.31, .47] |
| Direct effect | .13[a] | .05 | < .01 | [.04, .23] |
| Total indirect effect | .26[a] | .04 | < .001 | [.20, .34] |
| FTP→AB→AA | .23[a] | .04 | < .001 | [.16, .33] |
| FTP→AE→AA | .03[a] | .02 | < .05 | [.003, .07] |

*Note*. FTP = future time perspective, AA = academic achievement, AB = academic burnout, AE = academic achievement; [a] Empirical 95% confidence interval exclude zero.

academic engagement was much smaller compared to academic burnout, indicating that the effect of FTP on improving academic achievement is more significant through reducing academic burnout than through enhancing academic engagement.

As shown in Fig 2, future time perspective (FTP) influences academic achievement through two pathways. First, FTP significantly negatively predicts academic burnout (β = -0.44, p < 0.001, 95% CI [-0.91, -0.59]), and in turn, academic burnout significantly negatively predicts academic achievement (β = -0.52, p < 0.001, 95% CI [-0.48, -0.19]). In the second pathway, FTP significantly positively predicts academic engagement (β = 0.32, p < 0.01, 95% CI [0.97, 1.47]), and academic engagement subsequently significantly positively predicts academic achievement (β = 0.11, p < 0.05, 95% CI [0.01, 0.09]).

## Discussion

Academic institutions are increasingly focused on improving academic achievement to meet global demands and prepare undergraduates for their roles as global citizens [1, 3]. Within this context, this study addresses a critical issue: whether future time perspective (FTP) influences

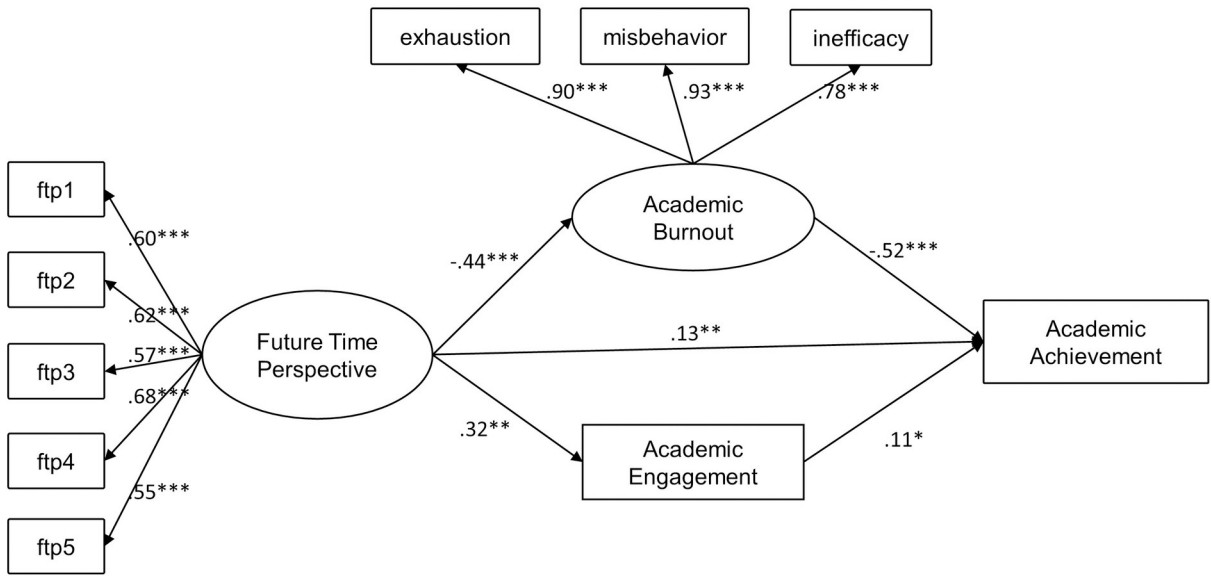

**Fig 2. Mediation analyses of future time perspective on academic achievement via academic burnout and engagement.** *Notes*. *p < .05; **p < .01; ***p < .001.

academic achievement among undergraduates in the context of Chinese higher education, and what mechanisms underlie this influence. Using a cross-sectional quantitative research design, we investigated the impact of FTP on academic achievement and explored the mediating roles of two distinct aspects of academic well-being–academic burnout and academic engagement.

Consistent with theoretical frameworks and previous research [4, 5, 23, 24, 53], this present study supports that FTP is positively related to academic achievement and serves as a positive predictor of academic achievement. From the lens of time perspective theory, individuals with high FTP focus on future-oriented, long-term goals [5, 16, 18], which aligns well with the nature of academic pursuits. Additionally, individuals with high FTP demonstrate better self-control, rational thinking [20], persistence in studying [17], effective learning strategies [26] and learning autonomy [28]. Therefore, a strong FTP leads to improved academic outcomes.

This study also supports the mediating roles of academic burnout and academic engagement between future time perspective (FTP) and academic achievement. As two aspects of academic well-being [31], academic burnout and academic engagement serve as negative and positive emotional mediators [30], respectively, influencing the impact of FTP on academic achievement. Our findings highlight that FTP influences academic achievement through both positive and negative emotional channels.

On one hand, FTP enhances academic achievement by promoting academic engagement. Consistent with previous studies [5, 9, 27, 31], FTP positively predicts academic engagement, which in turn positively predicts academic achievement. Individuals with a high FTP tend to have higher expectations for future goals and outcomes and invest more effort in their academic pursuits, thereby leading to improved academic performance. From a self-determination theory perspective, students with a high FTP may experience greater intrinsic motivation, as they perceive their academic efforts as more aligned with long-term goals [54]. This sense of purpose enhances their engagement in learning activities, which is a key determinant of academic success. In this way, engagement is not just an emotional state but also a manifestation of the underlying motivation driven by FTP [4]. Furthermore, engagement reflects a high level of autonomy and competence–two key components of SDT that promote adaptive learning behaviors [55]. When students are motivated by a strong FTP, they are more likely to seek out challenges, persist in the face of difficulties, and adopt effective learning strategies, all of which contribute to better academic outcomes. Therefore, FTP fosters engagement by enhancing students' intrinsic motivation, which directly impacts academic achievement.

On the other hand, FTP mitigates academic burnout, thereby enhancing academic achievement. While previous research has not directly explored the relationship between FTP and academic burnout, there is substantial evidence indicating a negative correlation between FTP and other negative academic emotions, such as procrastination [33] and disaffection [9]. The motivational function of FTP [56] may help regulate negative emotions. Individuals with a future-oriented perspective are likely better at managing and regulating negative emotions while planning and progressing towards their goals, thereby reducing the risk of burnout and improving academic outcomes. From an SDT perspective, academic burnout can be seen as a form of frustration of basic psychological needs, particularly the needs for autonomy and competence [57]. When students lack of a strong FTP, they may experience feelings of helplessness or hopelessness about their academic future, which could lead to academic burnout. In contrast, students with a high FTP are more likely to engage in proactive coping strategies, such as setting clear goals and managing time effectively, which helps protect against burnout [4].

This study expands the understanding of FTP by highlighting its role in mitigating negative academic emotions like academic burnout. In summary, the findings suggest that individuals with FTP achieve better academic outcomes by increasing their engagement and effectively managing their burnout.

## Conclusion and implications

The present study confirmed the relationship between future time perspective (FTP) and academic achievement, expanding the understanding of related mediating mechanisms and further supporting time perspective theory. In the context of Chinese higher education, undergraduate students with a strong FTP orientation are more likely to achieve higher academic achievement. FTP not only improves academic engagement, but also reduces academic burnout, both of which, in turn, enhance academic achievement. Notably, academic burnout had a greater mediating effect than academic engagement in the relationship between FTP and academic achievement. This study emphasizes the important role of FTP and its mediators in promoting academic success.

This study makes several contributions to the literature. First, by employing a cross-sectional quantitative research design, it enriches the understanding of the relationship between future time perspective (FTP) and academic achievement in a non-Western cultural context. Second, it extends this relationship by incorporating academic burnout and academic engagement as mediating mechanisms, demonstrating that FTP influences academic achievement through both enhancing engagement and mitigating burnout. Third, building on prior research linking FTP with negative academic emotions such as procrastination and disaffection, this study is the first to investigate the relationship between FTP and academic burnout, thereby expanding the potential role of FTP in managing negative emotions. Fourth, the findings have practical implications. Previous research has shown that time perspective can be shaped through interventions [58, 59]. Based on the positive effects of FTP on academic emotions and outcomes found in this study, interventions aimed at developing or shaping individuals' FTP could be designed and implemented to improve academic emotions and enhance academic achievement.

Despite these contributions, this study has several limitations. First, while both academic engagement and burnout mediated the relationship between FTP and academic achievement, the mediating effect of engagement was much smaller compared to burnout. This suggests the need for further research to explore the potential differential roles of FTP in regulating positive and negative emotions. Second, the cross-sectional design limits the ability to establish causality, and the reliance on self-reported data (except official GPA scores) may introduce response bias [60]. The gender imbalance of the sample (with 71.1% female) also raises concerns for the generalizability of the findings. Future research can use more representative sampling methods and longitudinal designs to provide more robust results. Qualitative methods could also be included for deeper insights into students' psychological states and behaviours. Third, in the context of the four-year undergraduate system in Chinese higher education, this study did not differentiate between academic years. Future research should examine whether the relationships between the variables differ across academic years, with particular attention to the transitional first year and the final year before graduation. Lastly, it is important to note that this study focused solely on individual factors influencing academic achievement and did not consider contextual factors, such as teacher support. Research by Liu et al. [61] and Li et al. [62] suggests that teacher support is a significant predictor of academic buoyancy, which can, in turn, influence academic outcomes. Future research could examine the interaction between individual factors, such as future time perspective (FTP), and contextual factors, such as teacher support, to better understand how these factors jointly shape students' learning experiences and academic outcomes.

## Supporting information

**S1 Appendix. Zimbardo future time perspective scale.**
(DOCX)

**S2 Appendix. Academic burnout scale.**
(DOCX)

**S3 Appendix. Academic engagement scale.**
(DOCX)

**S4 Appendix. Summary of factor loading, goodness-of-fit indices, and reliability for measurement scales.**
(DOCX)

**S1 Dataset.**
(RAR)

## Acknowledgments

I would like to express my sincere gratitude to all the students who participated in this study and to Prof. Zahyah Hanafi and Prof. Nordin Abd Razak for their invaluable guidance and constructive feedback, which greatly enhanced the quality of this manuscript.

## Author Contributions

**Conceptualization:** Shuai Hong.

**Data curation:** Shuai Hong.

**Formal analysis:** Shuai Hong.

**Investigation:** Shuai Hong.

**Methodology:** Shuai Hong.

**Resources:** Shuai Hong.

**Software:** Shuai Hong.

**Validation:** Shuai Hong.

**Writing – original draft:** Shuai Hong.

**Writing – review & editing:** Shuai Hong.

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
