## [Decision Letter · Decision Letter 0]

8 Oct 2024

PONE-D-24-35823The Impact of Future Time Perspective on Academic Achievement: Mediating Roles of Academic Burnout and EngagementPLOS ONE

Dear Dr. Hong,

Thank you for submitting your manuscript to PLOS ONE. After careful consideration, we feel that it has merit but does not fully meet PLOS ONE’s publication criteria as it currently stands. Therefore, we invite you to submit a revised version of the manuscript that addresses the points raised during the review process.

We look forward to receiving your revised manuscript.

Kind regards,

Leona Cilar Budler

Academic Editor

PLOS ONE

Journal Requirements:

2. We note that you have indicated that there are restrictions to data sharing for this study. PLOS only allows data to be available upon request if there are legal or ethical restrictions on sharing data publicly. For more information on unacceptable data access restrictions, please see http://journals.plos.org/plosone/s/data-availability#loc-unacceptable-data-access-restrictions. Before we proceed with your manuscript, please address the following prompts: a) If there are ethical or legal restrictions on sharing a de-identified data set, please explain them in detail (e.g., data contain potentially identifying or sensitive patient information, data are owned by a third-party organization, etc.) and who has imposed them (e.g., a Research Ethics Committee or Institutional Review Board, etc.). Please also provide contact information for a data access committee, ethics committee, or other institutional body to which data requests may be sent. b) If there are no restrictions, please upload the minimal anonymized data set necessary to replicate your study findings to a stable, public repository and provide us with the relevant URLs, DOIs, or accession numbers. For a list of recommended repositories, please see https://journals.plos.org/plosone/s/recommended-repositories. You also have the option of uploading the data as Supporting Information files, but we would recommend depositing data directly to a data repository if possible. We will update your Data Availability statement on your behalf to reflect the information you provide.

Additional Editor Comments:

There are some major issues listed by reviewers. Please, read all comments and suggestions carefully to improve paper quality. Also, check all journal guidelines to prepare your paper.

Reviewers' comments:

Reviewer's Responses to Questions

**Comments to the Author**

1. Is the manuscript technically sound, and do the data support the conclusions?

Reviewer #1: Yes

Reviewer #2: Partly

2. Has the statistical analysis been performed appropriately and rigorously? 

Reviewer #1: Yes

Reviewer #2: No

3. Have the authors made all data underlying the findings in their manuscript fully available?

Reviewer #1: Yes

Reviewer #2: Yes

4. Is the manuscript presented in an intelligible fashion and written in standard English?

Reviewer #1: Yes

Reviewer #2: No

5. Review Comments to the Author

Reviewer #1: Please Update your references and new one.

Provide items of your questionnaire in appendix or in one table.

Please provide one table for reporting Standardized factor loadings range, goodness of fit, and reliability for the factors of the instruments.

In introduction, please mention data about frequency of these 3 concepts in your academic context.

Data collected from Which country? Malaysia or china?

If china, you have a specific scale for university student engagement, why you chose the Chinese adaptation of the Utrecht Work Engagement Scale for Student (UWES-S, Li & Huang, 2010).

“Psychometric properties of the university student engagement inventory among Chinese students”

Reviewer #2: The topic of the paper is of significance, and the research is conducted following a quantitative approach with some theoretical basis in data processing. However, there are certain issues that need to be addressed:

1. Insufficient Theoretical Basis for the Research Topic: The rationale for how Future Time Perspective (FTP) affects academic achievement through burnout and engagement is not sufficiently supported, particularly in terms of theoretical backing. The author mentions FTP and Self-Determination Theory (SDT) but does not adequately address how these theories relate to the paper. The author should refer to the following literature to substantiate the rationale for the research topic:

o Liu, H., Zhong, Y., Chen, H., & Wang, Y. (2023). The mediating roles of resilience and motivation in the relationship between students’ English learning burnout on engagement: A conservation-of-resources perspective. International Review of Applied Linguistics in Language Teaching. https://doi.org/10.1515/iral-2023-0089

2. Missing Relationship between Burnout and Engagement in the Model: Although the author does not explicitly address the relationship between burnout and engagement in the established model, there is a known connection between these two factors, as evidenced in previous literature. The author should provide reasons for not establishing this link and should refer to the following literature:

o Liu, H., Zhong, Y., Chen, H., & Wang, Y. (2023). The mediating roles of resilience and motivation in the relationship between students’ English learning burnout on engagement: A conservation-of-resources perspective. International Review of Applied Linguistics in Language Teaching. https://doi.org/10.1515/iral-2023-0089.

o Liu, H. & Zhong, Y. (2022). English learning burnout: Scale validation in the Chinese context. Frontiers in Psychology. 13:1054356, 1-10. https://doi.org/ 10.3389/fpsyg.2022.1054356.

o Liu, H., Li. J., & Fang. F. (2022). Examining the complexity between boredom and engagement in English Learning: Evidence from Chinese high school students. Sustainability, 14, 16920: 1-12. https://doi.org/10.3390/su142416920.

o Wang, X. & Liu, H. (2024). Exploring the moderation roles of emotions, attitudes, environment, and teachers on the impact of motivation on learning behaviors in students’ English learning. Psychological Reports, online publication, 1-27. https://doi.org/10.1177/00332941241231714

o Wang, Y. & Liu. H. (2022). The mediating roles of boredom and buoyancy in the relationship between autonomous motivation and engagement among Chinese senior high school EFL learners. Frontiers in Psychology. 13:992279, 1-12. https://doi.org/10.3389/fpsyg.2022.992279

3. Lack of Depth in Discussion: The discussion section of the paper is not sufficiently in-depth. Each paragraph in the discussion section essentially reiterates the research findings, but a more thorough discussion is needed, particularly regarding the mediating effects.

4. Inconsistency in Formatting: For instance, on page 11, line 268, there is an in-text citation (Song & Feng, 2017) that does not adhere to the format used elsewhere in the paper.

5. Combining the Last Two Sections of the Paper: The conclusion and implications should be merged, with the major findings presented first (originally the conclusion section), followed by the paragraph starting at line 276, and ending with the paragraph starting at line 288.

6. The Importance of Teacher Support: The research suggests that teacher support is also an indispensable factor. The author should refer to the following literature and include it in the paper:

o Liu, H., Li, X., & Y. Yan (2023) Demystifying the predictive role of students’ perceived foreign language teacher support in foreign language anxiety: the mediating role of L2 grit. Journal of Multilingual and Multicultural Development. https://doi.org/10.1080/01434632.2023.2223171.

o Liu, H. & Li, X. (2023). Unravelling students’ perceived EFL teacher support. System. 115, 103048, 1-12. https://doi.org/10.1016/j.system.2023.103048.

o Li, X., Duan, S., & Liu, H. (2023). Unveiling the predictive effect of students’ perceived EFL teacher support on academic achievement: The mediating role of academic buoyancy. Sustainability, 15, 10205. 1-12. https://doi.org/10.3390/su151310205

The recommended articles above should be referred to and cited by the author, but the decision to cite them is independent of my review.

6. PLOS authors have the option to publish the peer review history of their article (what does this mean?). If published, this will include your full peer review and any attached files.

Reviewer #1: No

Reviewer #2: No

---

## [Author Response · Author response to Decision Letter 0]

12 Nov 2024

Ref: PONE-D-24-35823

The Impact of Future Time Perspective on Academic Achievement: Mediating Roles of Academic Burnout and Engagement

Dear Editors and Reviewers,

We sincerely appreciate the time and effort you have dedicated to reviewing our manuscript entitled “The Impact of Future Time Perspective on Academic Achievement: Mediating Roles of Academic Burnout and Engagement”. We are grateful for the insightful feedback provided by the reviewers, and we have carefully addressed each of the comments in our revisions, as outlined below. 

Reviewers' comments to the Author: 

Reviewer #1:

Comment 1. Please Update your references and new one.

Response: 

Thank you for your comments on the references. We have reviewed and updated the references in the manuscript to include the most recent literature relevant to our study. While we have made every effort to include major new studies, we recognize that there may be additional relevant literature to consider in future updates. We have also ensured a consistent citation style throughout the manuscript.

Comment 2. Provide items of your questionnaire in appendix or in one table.

Response: 

Thank you for your suggestion. We have included all questionnaire items in the Supplemental Material for clarity and transparency. Specifically:

- Appendix A includes items from the Zimbardo Future Time Perspective Scale.

- Appendix B provides the items from the Academic Burnout Scale.

- Appendix C presents the items from the Academic Engagement Scale.

Please note that the questionnaires were administered in Chinese, the participants' local language. For the purposes of publication and accessibility to a broader audience, we have translated the items into English in the supplemental materials.

Comment 3. Please provide one table for reporting Standardized factor loadings range, goodness of fit, and reliability for the factors of the instruments.

Response: 

Thank you for your valuable suggestions. In the revised manuscript, we have added a table in the Supplementary Materials (Appendix D. Summary of Factor Loading, Goodness-of-Fit Indices, and Reliability for Measurement Scales) that describes the standardized factor loading ranges, goodness-of-fit indices, and reliabilities for each of the factors in the instruments used in this study (Future Time Perspective, Academic Burnout, and Academic Engagement). While the Cronbach's alpha and CFA goodness-of-fit results are described in the Methods - Measures section, we have now combined these details and factor loading ranges into the table based on your suggestions to provide a more complete picture of the quality and reliability of the measurement models.

Comment 4. In introduction, please mention data about frequency of these 3 concepts in your academic context.

Response:

Thank you for your suggestion, which has helped us strengthen the manuscript.

To address the request for data on the prevalence of the three core variables, we have added recent statistics on the occurrence of academic burnout and academic engagement, among university students, particularly within the context of Chinese higher education. The revised content includes:

“Academic burnout is a pervasive issue among university students, particularly in high-pressure academic environments like those in China. Research highlights a significant prevalence of this problem: a recent cross-sectional study involving 22,983 Chinese university students found that 59.9% experienced academic burnout [37]. Similarly, Hong and Hanafi report an upward trend in academic burnout rates among Chinese university students, with prevalence rising from 37.5% in 2007 to 61.63% in 2019 [38]. Other studies underscore that academic burnout, alongside internet addiction and loneliness, is a prevalent challenge among Chinese college students and is often linked with heightened stress levels [39]. This high incidence of burnout impacts students’ academic performance and underscores the value of academic engagement as a protective factor and a pathway to enhanced academic success.”

Regarding future time perspective (FTP), there is no incidence rate, as it represents a personality trait rather than a condition with a measurable prevalence. Instead, FTP has gained substantial attention in educational psychology as a research focus due to its strong connections with academic motivation, goal-setting, and self-regulation. Recent studies increasingly identify FTP as a key predictor of academic engagement and achievement, positioning it as a critical factor for understanding student succuss in high-stakes academic environments.

Comment 5. Data collected from Which country? Malaysia or china?

If China, you have a specific scale for university student engagement, why you chose the Chinese adaptation of the Utrecht Work Engagement Scale for Student (UWES-S, Li & Huang, 2010).

“Psychometric properties of the university student engagement inventory among Chinese students”

Response: 

Thank you for your insightful question regarding our choice of measurement tools. We clarify below the rationale behind our selection of the Chinese adaptation of the Utrecht Work Engagement Scale for Students (UWES-S, Li & Huang, 2010) for this study, which collected data in mainland China.

Our rationale has two primary considerations:

First, from the theoretical basis and conceptualization of academic engagement, the UWES-S is aligned with our operational definition of academic engagement as a positive, fulfilling, motivational state characterized by vigor, dedication, and absorption. This definition follows the conceptualization by Schaufeli and colleagues (Schaufeli, Martínez, et al., 2002; Schaufeli, Salanova, et al., 2002), who developed the concept of academic engagement as the positive counterpart to academic burnout, thus focusing on emotional well-being in academic settings. From this perspective, engagement is seen as a psychological state that reflects a student’s vitality and dedication to their studies, contrasting academic burnout and contributing to the broader construct of academic well-being. The construct matches our study’s model, which emphasizes engagement as a counterbalance to burnout.

Alternatively, some researchers, notably Fredricks and colleagues (Fredricks et al., 2004), propose a more comprehensive model of engagement that includes behavioral, emotional, and cognitive dimensions. This model has been adopted by Chinese researchers, including Zhang (2012) and Liao (2011), who developed the locally specific University Student Engagement Inventory to capture engagement in these three dimensions. However, this broader conceptualization differs from our study’s focus, which aligns more closely with the affective, motivational aspects of engagement as defined in the UWES-S framework. 

Second, from the perspective of the prevalence and established use of the UWES-S in China. The UWES-S has been widely validated and extensively used in Chinese academic research as a standard tool for assessing academic engagement. It remains one of the most frequently applied instruments for this construct, both in high-impact Chinese academic journals and on CNKI (China National Knowledge Infrastructure), the leading Chinese academic search engine. For instance, the Chinese version of the UWES-S, first validated by Fang et al. (2008), has been cited 1,358 times. The subsequent revision for university student populations by Li and Huang (2010) has been cited over 593 times. Additionally, recent influential publications continue to use the UWES-S as a preferred instrument for measuring academic engagement in Chinese university settings (e.g., LEI & YIN, 2024; PAN et al., 2023).

In sum, the UWES-S was chosen as it aligns with our conceptual framework and operational definition of academic engagement and is also well-supported in the Chinese academic context, where it has a high degree of validation and established use. We appreciate the opportunity to elaborate on these considerations and hope this provides sufficient justification for our choice.

Reviewer #2: The topic of the paper is of significance, and the research is conducted following a quantitative approach with some theoretical basis in data processing. However, there are certain issues that need to be addressed:

Comment 1. Insufficient Theoretical Basis for the Research Topic: The rationale for how Future Time Perspective (FTP) affects academic achievement through burnout and engagement is not sufficiently supported, particularly in terms of theoretical backing. The author mentions FTP and Self-Determination Theory (SDT) but does not adequately address how these theories relate to the paper. The author should refer to the following literature to substantiate the rationale for the research topic:

o Liu, H., Zhong, Y., Chen, H., & Wang, Y. (2023). The mediating roles of resilience and motivation in the relationship between students’ English learning burnout on engagement: A conservation-of-resources perspective. International Review of Applied Linguistics in Language Teaching. https://doi.org/10.1515/iral-2023-0089

Response: 

Thank you for your valuable feedback and for highlighting the importance of strengthening the theoretical foundation of the paper. We recognize the need to further clarify the relationship between Future Time Perspective (FTP), academic achievement, academic burnout, and engagement, particularly within the framework of Self-Determination Theory (SDT).

In the revised manuscript, we have expanded the theoretical section to provide a more comprehensive explanation of how FTP influences academic achievement through academic burnout and engagement. Specifically, we have elaborated on how FTP aligns with the principles of SDT, particularly in terms of how an individual’s future-oriented goals can enhance intrinsic motivation, which in turn impacts both engagement and burnout. FTP promotes a sense of purpose and direction, helping students engaged in their academic activities while reducing feeling of burnout.

Additionally, we have integrated the recommended literature into our theoretical section. The study by Liu et al. (2023) contributes valuable insights on the mediating roles of resilience and motivation in academic burnout and engagement, and we have cited this work to further substantiate our rationale. This reference enhances our argument that personal resources such as motivation and resilience, as influenced by FTP, are critical in shaping students’ engagement and mitigating burnout.

We believe these additions will clarify the theoretical grounding of our research and make the relationships between FTP, academic burnout, engagement, and achievement more explicit.

Comment 2. Missing Relationship between Burnout and Engagement in the Model: Although the author does not explicitly address the relationship between burnout and engagement in the established model, there is a known connection between these two factors, as evidenced in previous literature. The author should provide reasons for not establishing this link and should refer to the following literature:

o Liu, H., Zhong, Y., Chen, H., & Wang, Y. (2023). The mediating roles of resilience and motivation in the relationship between students’ English learning burnout on engagement: A conservation-of-resources perspective. International Review of Applied Linguistics in Language Teaching. https://doi.org/10.1515/iral-2023-0089.

o Liu, H. & Zhong, Y. (2022). English learning burnout: Scale validation in the Chinese context. Frontiers in Psychology. 13:1054356, 1-10. https://doi.org/ 10.3389/fpsyg.2022.1054356.

o Liu, H., Li. J., & Fang. F. (2022). Examining the complexity between boredom and engagement in English Learning: Evidence from Chinese high school students. Sustainability, 14, 16920: 1-12. https://doi.org/10.3390/su142416920.

o Wang, X. & Liu, H. (2024). Exploring the moderation roles of emotions, attitudes, environment, and teachers on the impact of motivation on learning behaviors in students’ English learning. Psychological Reports, online publication, 1-27. https://doi.org/10.1177/00332941241231714

o Wang, Y. & Liu. H. (2022). The mediating roles of boredom and buoyancy in the relationship between autonomous motivation and engagement among Chinese senior high school EFL learners. Frontiers in Psychology. 13:992279, 1-12. https://doi.org/10.3389/fpsyg.2022.992279

Response: 

Thank you for your insightful comment regarding the relationship between burnout and engagement. We appreciate your concern about the absence of direct exploration of this relationship in our research model, and we would like to address this point in more detail.

While we did not explicitly examine the relationship between academic burnout and engagement in our model, we focused on the impact of future time perspective (FTP) on academic achievement, with academic burnout and engagement serving as two distinct mediators in separate pathways, each reflecting different aspects of students’ academic experience. Previous studies have shown that burnout and engagement are negatively correlated, and some research – particularly in areas like language learning – suggests that burnout can negatively predict engagement. However, we conceptualized these two constructs as opposite yet distinct phenomena. This approach aligns with the broader literature, which treats burnout and engagement as representing two ends of the academic emotional spectrum. In our model, we emphasize this opposition to explain how FTP influences academic outcomes through two contrasting pathways: one where burnout depletes personal resources and another where engagement activates motivation and effort.

Although we acknowledge the potential for a negative relationship between burnout and engagement, the primary focus of our current research was not to exploring their interaction but on treating them as independent mediators in the relationship between FTP and academic achievement. In light of the literature you suggested, we have revised the manuscript to clarify why this relationship was not addressed in our model. We have also cited the relevant studies to explain the theoretical basis for this decision. Future research could further investigate the more complex relationship between these two constructs, particularly in diverse populations, to gain a better understand of their dynamic interaction.

We appreciate your suggestion and will consider this in our future work, where we aim to explore the more nuanced relationships between burnout and engagement across different contexts.

Comment 3. Lack of Depth in Discussion: The discussion section of the paper is not sufficiently in-depth. Each paragraph in the discussion section essentially reiterates the research findings, but a more thorough discussion is needed, particularly regarding the mediating effects.

Response: 

Thank you for your valuable feedback. We have revised the Discussion section to provide a more in-depth analysis, particularly regarding the mediating effects of academic burnout and academic engagement. In the revised manuscript, we have expanded on the theoretical mechanisms underlying these mediating effects by integrating Self-Determination Theory (SDT) to explain how FTP influences both academic burnout and engagement, and how these factors, in turn, impact academic achievement. We believe these additions enhance the depth and theoretical grounding of the discussion.

We appreciate your constructive suggestion and hope the revisions address your concerns.

Comment 4. Inconsistency in Formatting: For instance, on page 11, line 268, there is an in-text citation (Song & Feng, 2017) that does not adhere to the format used elsewhere in the paper.

Response: 

Thank you for pointing out the inconsistency in formatting. We have carefully reviewed the manuscript and corrected the in-text citation (Song & Feng, 2017) on line 86 and line 300, to match the format used consistently throughout the paper. Additionally, we conducted a thorough review of the entire manuscript and corrected any other formatting inconsistencies in citations to ensure uniformity. We appreciate your attention to detail.

Comment 5. Combining the Last Two Sectio

---

## [Decision Letter · Decision Letter 1]

17 Dec 2024

The Impact of Future Time Perspective on Academic Achievement: Mediating Roles of Academic Burnout and Engagement

PONE-D-24-35823R1

Dear Dr. Hong,

We’re pleased to inform you that your manuscript has been judged scientifically suitable for publication and will be formally accepted for publication once it meets all outstanding technical requirements.

Kind regards,

Leona Cilar Budler

Academic Editor

PLOS ONE

Additional Editor Comments (optional):

Major issues have been resolved.

Reviewers' comments:

Reviewer's Responses to Questions

**Comments to the Author**

1. If the authors have adequately addressed your comments raised in a previous round of review and you feel that this manuscript is now acceptable for publication, you may indicate that here to bypass the “Comments to the Author” section, enter your conflict of interest statement in the “Confidential to Editor” section, and submit your "Accept" recommendation.

Reviewer #2: All comments have been addressed

2. Is the manuscript technically sound, and do the data support the conclusions?

Reviewer #2: Yes

3. Has the statistical analysis been performed appropriately and rigorously? 

Reviewer #2: Yes

4. Have the authors made all data underlying the findings in their manuscript fully available?

Reviewer #2: Yes

5. Is the manuscript presented in an intelligible fashion and written in standard English?

Reviewer #2: Yes

6. Review Comments to the Author

Reviewer #2: (No Response)

7. PLOS authors have the option to publish the peer review history of their article (what does this mean?). If published, this will include your full peer review and any attached files.

Reviewer #2: No

---

## [Editor Report · Acceptance letter]

23 Jan 2025

PONE-D-24-35823R1 

PLOS ONE

Dear Dr. Hong, 

I'm pleased to inform you that your manuscript has been deemed suitable for publication in PLOS ONE. Congratulations! Your manuscript is now being handed over to our production team.

Kind regards, 

on behalf of

Dr. Leona Cilar Budler 

Academic Editor

PLOS ONE